# Experiment and Numerical Simulation on Thermal Cycling Performance of YSZ-Based Sealing Coatings with "Brick-Mud" Layered Structure

Taotao Cheng [1,2,*], Yuelu Dong [2], Liang Ma [2,*], Zhibing Wu [2], Jun Wang [3], Xiang Ma [4], Zhiping Wang [1,2,*] and Shijie Dai [1,*]

1   School of Mechanical Engineering, Hebei University of Technology, No. 5340, Xiping Road, Beichen District, Tianjin 300401, China
2   Tianjin Key Laboratory for Civil Aircraft Airworthiness and Maintenance, Civil Aviation University of China, No. 2898, Jinbei Road, Dongli District, Tianjin 300300, China; yueludong@outlook.com (Y.D.); 15972920553@163.com (Z.W.)
3   State Key Laboratory of Tribology, Department of Mechanical Engineering, Tsinghua University, No. 30, Shuangqing Road, Haidian District, Beijing 100084, China; wjun@tju.edu.cn
4   Engine Overhaul Department, Aircraft Maintenance and Engineering Corporation Co., Ltd. (AMECO), No. 2, Capital Airport Road, Chaoyang District, Beijing 100621, China; maxiang@ameco.com.cn
*   Correspondence: ttcheng@cauc.edu.cn (T.C.); mliang@tju.edu.cn (L.M.); zpwang@cauc.edu.cn (Z.W.); dsj@hebut.edu.cn (S.D.)

**Abstract:** The failure of premature thermal cycling spalling off is the bottleneck problem currently faced by yttrium oxide partially stabilized zirconia (YSZ) ceramic-based sealing coatings. Studies on the thermal cycling performance of coatings with "brick-mud" structures were carried out by experimental and simulation methods in this paper. The results showed that, as the thickness of "mud" layer increased, the bonding strength of the "brick-mud" structure coatings gradually decreased. When the thickness of the "mud" layer was about 3 μm and 10 μm, the thermal cycling lives of the T1 and T2 coatings were improved by 90.0% and 135.7%, respectively, compared with conventional coating (T0 coating), while that of the T3 coating (containing thick "mud" layers of about 20 μm) was decreased by 81.4%. The stress field of M2 "mud" layers with different thicknesses was subjected to a comprehensive effect by thermal mismatch stress and pores in "brick" layer. Compared with the medium and thick "mud" layers, the thin "mud" layer sustained obvious larger $\sigma_{22\ max}$ and $\sigma_{12\ max}$, indicating its potential for the preferential initiation of transverse microcracks. In addition, the thin "mud" layer withstood the largest $\sigma_{11\ max}$ and had the strongest potential for longitudinal crack growth. Both transverse and longitudinal cracking could consume energy during thermal cycling and reduce the stress concentration at the top coating/bond coating interface. These were the main reasons for the improvements in the thermal cycling performances of the T1 and T2 coatings. The degree of crack deflection and the capacity of energy dissipation in the "mud" layer increased significantly with its thickness. However, the propagation length of transverse cracks also gradually increased in the meantime. Especially when the "mud" layer was 20 μm, the length of the transverse cracks increased rapidly. Thus, early interlayer delamination failure occurred in the T3 coating during thermal cycling.

**Keywords:** "brick-mud" structure; thermal cycling performance; internal stress; crack propagation; interlayer delamination

## 1. Introduction

As the thrust–weight ratio of advanced engines continues to increase, the temperature in front of the turbine also continues to rise, resulting in the airflow temperature at the turbine shroud exceeding 1000 °C. The commonly used alloy-based sealing coatings can no longer meet these service requirements [1,2]. Consequently, ceramic-based sealing

coatings have become one of the current research hotspots, and are urgent needed for high-performance engines [3–5]. The structure of ceramic-based sealing coatings is generally a complex multi-layer composite system, including a Ni-based superalloy substrate, a bonding coating (BC coating) providing transitional properties, a top coating (TC coating) offering thermal insulation and abradable performance, and a thermally grown oxide layer (TGO) between the BC and TC coatings.

While increasing the temperature of heat resistance, ceramic-based sealing coatings are mainly facing the following two challenges. The first is the low deposition efficiency caused by the high melting point of ceramic materials. The second is the poor thermal cycling performance caused by the large thickness of the TC coating and the thermal cycling load from over 1000 °C to room temperature. In response to the first challenge, the author of this paper has significantly improved the deposition efficiency of the TC coating by doping a YAG bonding phase into the agglomerated particles [6].

In response to the second challenge, researchers usually improve the thermal cycling performance by optimizing the original continuous stacking structure of conventional ceramic coatings. However, the methods of releasing thermal cycling internal stress through "columnar" or "quasi columnar" structures usually face challenges such as a high cost, difficult preparation, or low molding efficiency [7–11]. The methods of releasing thermal cycling internal stress through micro cracking generally have limitations such as uncontrollable lateral cracking or poor structural uniformity and stability [12,13]. At present, the problem of premature thermal cycling spalling remains a bottleneck issue for ceramic coatings [12,14]. In recent years, studies on the preparation or optimization of ceramic coatings by using resins have attracted widespread attention. In 2007 [15], P Ctibor et al. studied the effect of epoxy resin on the performance of alumina coatings. It was shown that the abrasion wear resistance of the epoxy resin sealed coatings was significantly better than that of the as-sprayed coatings. In 2011 [16], G Isgró et al. researched the deformation and strength of a dental ceramic following resin-cement coating. The results indicated that the resin-cement coating significantly increased the mean deflection and the mean bi-axial flexure strength for specimens against the uncoated state. In 2015 [17], P Luangtriratana et al. investigated the thermal barrier efficiency of five commercially available ceramic nano and micro particles deposited on the surfaces of glass fiber-reinforced epoxy composites (GRE). The results showed that the surface layers of all coated samples were uniform and there was strong adhesion between the coating and the substrate. Moreover, they did not adversely affect the mechanical properties of the GRE composites, while improving the mechanical property retention of the GRE composites after exposure to heat. In 2018 [18], W Deng et al. introduced epoxy resin (ER) into an as-sprayed 8 wt.% yttria-stabilized zirconia (8YSZ) coating by vacuum impregnation to prepare an 8YSZ-ER coating. It turned out that the hardness, toughness, cohesive strength, density, and cavitation performance of the 8YSZ-ER coating were greatly improved. In 2019 [19], Y Y Wang et al. prepared silica/epoxy hybrid polymers as sealing layers on ceramic coatings and studied their stability upon thermal treatment. When the hybrids were brushed on ceramic coatings, they infiltrated to a depth of around 60 μm. As sealing layers, they probably experienced partially decomposition and slumping, and then filled the internal defects of the ceramic coatings. In 2022 [20], a novel epoxy-based ablative-resistant coating was developed using modified EG, zinc borate, E-glass fibers, and epoxy resin by L F Hao et al. It was indicated that the coating was characterized by good heat insulation and heat-resistant parameters.

The above research results demonstrated that specific resins can improve the microstructure, heat resistance, and mechanical properties of ceramic coatings, which provided great inspiration for the authors of this paper. In the previous research of the authors [21,22], principles inspired by bionic seashells were applied to the structural optimization of the TC coating, and a novel "brick-mud (resin)"-structured YSZ ceramic-based sealing coating was constructed. During the thermal cycling process, mechanisms such as micro cracking, crack branching, and crack deflection occurred in the novel structure TC coating, prolonging the crack propagation path and consuming part of the internal

stress, thereby improving the thermal cycling life of the coating to a certain extent. However, the evolution rules of internal stress during thermal cycling and the mechanism of crack initiation and propagation for the new structure coating have not been deeply explored. In this paper, the influence of "mud" layer thickness on the thermal cycling performance was researched through experimental methods and simulation calculations. Finally, the thermal cycling failure mechanism of the "brick-mud" layered structure coating was further elucidated.

## 2. Materials and Methods

### 2.1. Experimental Materials and Methods

2.1.1. Raw Materials of the "Brick" Layer and the "Mud" Layer

(1)  Main components of the "brick" layer

A YSZ-based porous coating was prepared as the "brick" layer (YSZ "brick" layer for short) [18]. The YSZ raw powders (ST-O-006-2, Shanghai Shuitian Material Technology Co., Ltd., Shanghai, China) and the Poly-p-hydroxybenzoate (PHB) powders (CGZ-351-2, Zhonghao Chenguang Chemical Research Institute Co., Ltd., Zhonghao, China) were used for spray granulation in this work.

(2)  Raw materials of the "mud" layer

A methyl silicone resin (SILRES MK) high-temperature adhesive was selected as the "mud" layer in the "brick-mud" composite unit. The MK resin powder was purchased from Wacker Chemie AG, München, Germany.

2.1.2. Preparation of the "Brick" Layer and the "Mud" Layer

(1)  Preparation of the "brick" layer

The raw materials of agglomerated powders used for the YSZ "brick" layer were crushed and mixed uniformly by a planetary ball mill according to Table 1. The preparation process of the agglomerated particles is shown in Table 2. The apparatus used for the APS process consisted of a controlling system (Praxair 3710, Praxair Surface Technologies, Indianapolis, IN, USA) and a spray gun (SG-100, Praxair Surface Technologies, Indianapolis, IN, USA). Helium gas was used as the plasma gas. The main processing parameters for the preparation of the YSZ "brick" layer are listed in Table 3.

**Table 1.** Processing parameters of the planetary ball mill.

| Temperature °C | Rotation Speed r·min$^{-1}$ | Milling Time min |
|---|---|---|
| 20~30 | 800 | 500 |

**Table 2.** Processing parameters for spray granulation.

| Inlet Temperature (°C) | Outlet Temperature (°C) | Slurry Flow Rate (mL min$^{-1}$) |
|---|---|---|
| 250 | 130 | 50 |

**Table 3.** Processing parameters of the APS.

| Voltage (V) | Current (A) | Spraying Distance (mm) | Powder Feeding Rate (g min$^{-1}$) |
|---|---|---|---|
| 40.5 | 800 | 125 | 15–20 |

(2)  Preparation of the "mud" layer

Firstly, the MK resin powder as the solute was added to isopropanol as the solvent, and then the MK resin high-temperature adhesive was obtained by mechanical stirring according to Table 4. Finally, the high-temperature adhesive was atomized and accelerated to be deposited on the sample through a high-viscosity pneumatic spray gun.

**Table 4.** Processing parameters of MK resin high-temperature adhesive.

| Weight of MK Powder (g) | Volume of Isopropanol (mL) | Rotation Speed (r·min$^{-1}$) | Stirring Time (min) | Temperature (°C) |
|---|---|---|---|---|
| 15 | 60 | 800 | 480 | 20–30 |

### 2.1.3. Design of "Brick-Mud" Layered Coatings Containing "Mud" Layers with Various Thicknesses

In order to investigate the effect of "mud" layers with different thicknesses on the thermal cycling performances of the "brick-mud" structure TC coatings, three sets of coatings were designed, as shown in Figure 1.

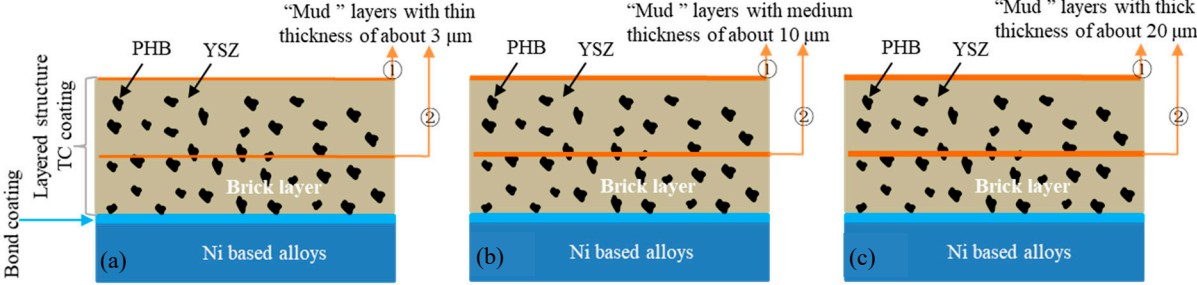

**Figure 1.** Structural schematic diagram of "brick-mud" layered coatings: (**a**) coating with thin "mud" layers; (**b**) coating with medium thickness "mud" layers; and (**c**) coating with thick "mud" layers.

### 2.1.4. Fabrication of "Mud" Layers with Various Thicknesses

A high-viscosity spray gun was used to prepare the "mud" layers with various thicknesses. When other process parameters such as the specification of the substrate (IN718), stress and distance remained unchanged, the thickness of the "mud" layers was mainly determined by the volume of the high-temperature adhesive sprayed. The preparation processes for "mud" layers with thin, medium, and thick thicknesses (labeled as T1, T2, and T3, respectively) are listed in Table 5.

**Table 5.** Main preparation process parameters of "mud" layers with three various thicknesses.

| Category | Specification of the Substrate (mm) | Stress (Mpa) | Distance (mm) | Volume of High-Temperature Adhesive Sprayed (mL) | Label |
|---|---|---|---|---|---|
| "Mud" layers with thin thickness | | 0.3 | 450 | 2 | T1 |
| "Mud" layers with medium thickness | $\Phi\ 25.4 \times 6$ | 0.3 | 450 | 5 | T2 |
| "Mud" layers with thick thickness | | 0.3 | 450 | 10 | T3 |

### 2.1.5. Bonding Strength Test

The bonding strength of the coatings was tested according to ASTM C633-13 [23], and the specimens were subjected to tensile testing by using the INSTRON 5982 (Instron Corporation, Boston, MA, USA) universal electronic material testing machine with a tensile speed of 1 mm/min. 2214 Regular Epoxy resin was selected as the adhesive. The tensile strength of the samples was calculated by the following equation:

$$S_\mathrm{m} = \frac{4F}{\pi d^2} \tag{1}$$

where $S_m$ is the tensile strength of the coating, $F$ is the maximum tensile load, and $d$ is the diameter of specimens with a value of 25.4 mm.

### 2.1.6. Thermal Cycling Test

The substrate of the thermal cycling specimens was IN 718 alloy with a specification of $\Phi$ 25.4 mm × 6 mm. The thermal cycling life was tested by self-developed thermal cycling equipment with automatic control functions for heating, heating preservation, and cooling processes, and was carried out according to the following steps: (1) the coating was heated to 1100 °C and kept for 10 min. (2) The coating was cooled for 5 min by compressed air. (3) The steps (1) and (2) were repeated and the coating was checked by visual inspection after each cooling process. Failure of the specimens was identified if the spalling area of the coating was more than 10% of the whole area.

### 2.1.7. Microstructural Characterization

The Labotom-5 high-speed cutting machine (diamond cutting wheel) was used to cut the coating samples with a "brick-mud" layered structure in a direction perpendicular to the coating surface. The thicknesses of the "mud" layers, microstructure, and chemical composition were characterized by optical microscopy (OM, OL4100, Olympus, Tokyo, Japan), scanning electron microscopy (SEM, Zeiss Gemini Sigma 300 VP, Oberkochen, Germany), and energy-dispersive spectrometry (EDS, attachment of SEM).

### *2.2. Numerical Model Development*

### 2.2.1. Geometry Model

ABAQUS-2021 finite element software was used to simulate and analyze the shell bionic "brick-mud" layered structure ceramic-based sealing coating. The structure and dimensions of the finite element model are shown in Figure 2. The model was designed as a dual-layer TC coating with varying thicknesses of "mud" layers. From top to bottom, there were two alternating layers of "mud" and "brick", a bonding layer (BC layer), and a substrate layer (SUB layer). In order to differentiate between the different positions of the "mud" layers, they were defined as Mud-1 (M1) layer, Brick-1 (B1) layer, Mud-2 (M2) layer, Brick-2 (B2) layer, BC layer, and SUB layer, ranging from the TC coating surface to the substrate layer.

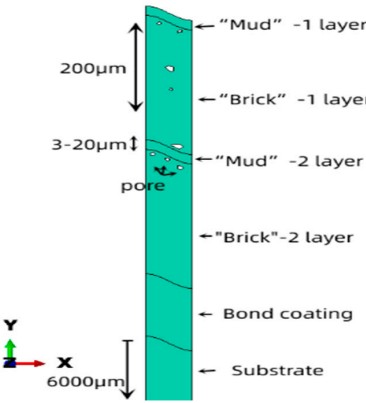

**Figure 2.** Finite element model of the "brick-mud" layered structure coating.

The "brick" layer was set to a thickness of 200 μm. To investigate the effect of the "mud" layer thickness on the stress field and crack propagation behavior, various thicknesses of the "mud" layer were utilized. The thicknesses of the thick "mud" layer ($M_T$ layer), medium thickness "mud" layer ($M_M$ layer), and thin "mud" layer ($M_t$ layer) were 20 μm, 10 μm, and 3 μm, respectively. In order to ensure the abradable performance of the TC coating, the "brick" layer in the TC coating usually contains a large number of pores with randomness. Consequently, this research designed some irregular pores with diameters of 5–15 μm in

the "brick" layer [24]. Additionally, the surface of the YSZ "brick" layer prepared by APS was rough and uneven, which can affect the stress field distribution and crack propagation behavior of the coating. Thus, the typical three-dimensional microscopic morphology of the YSZ "brick" layer surface, as shown in Figure 3a, was captured by using a laser confocal microscope. The surface height map and cross-sectional geometric profile obtained are shown in Figures 3b and 3c, respectively. The height parameters of the "brick" layer surface were measured by image analysis software of laser confocal microscopy and are presented in Table 6. The average values of the maximum surface height ($Sz$), maximum peak height ($Sp$), maximum valley depth ($Sv$), and arithmetic mean height ($Sa$) were 107.5 μm, 56.9 μm, 50.6 μm, and 11.6 μm, respectively. For the convenience of obtaining general rules, the interface was simplified as a cosine wave [25]. A point analysis was conducted on the profile to obtain the average value of peak-to-valley distances, and the designed cosine wave possessed a wavelength of 100 μm and an amplitude of 11.6 μm.

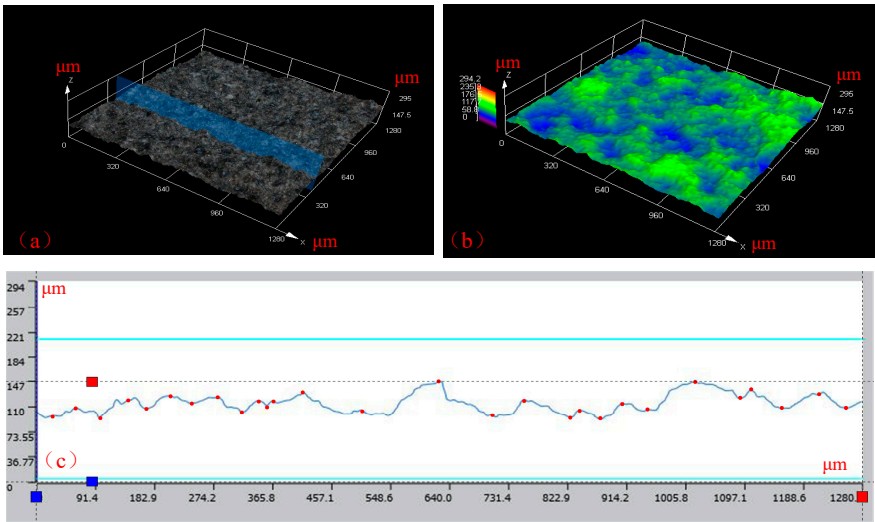

**Figure 3.** Three-dimensional microscopic morphology of the surface of YSZ "brick" layer: (**a**) location of surface contour line; (**b**) surface height map; and (**c**) surface contour line.

**Table 6.** Surface height parameters of YSZ "brick" layer measured by laser confocal microscopy.

| Sample No. | $Sz$/μm | $Sp$/μm | $Sv$/μm | $Sa$/μm |
|---|---|---|---|---|
| 1 | 111.545 | 59.015 | 52.530 | 11.433 |
| 2 | 96.655 | 49.568 | 47.087 | 10.745 |
| 3 | 114.213 | 61.976 | 52.237 | 12.651 |
| Average values | 107.471 | 56.853 | 50.618 | 11.609 |

### 2.2.2. Boundary Conditions and Thermal Loading

The boundary conditions for the numerical model are illustrated in Figure 4. In this study, the analysis was conducted based on a half-period model, which imposed periodic behavior on the entire model. The left side of the coating adopted symmetric boundary conditions, while the right side utilized periodic boundary conditions to ensure coordinated displacement in the *x*-axis direction [26]. The specific boundary conditions were as follows:

(1) The left boundary was constrained with symmetric constraints to prevent any displacement in the *X*-axis direction, while allowing free expansion in the *y*-axis direction.

(2) The bottom boundary was constrained to avoid rigid body displacement of the model in the *Y*-axis direction and allowed frictionless expansion in the *x*-axis direction.

(3) The right boundary was subjected to a coupled constraint, which ensured that it shared the same displacement as a specific point. This constraint guaranteed the overall displacement coordination of the right boundary.

(4)  The top boundary was left unconstrained to allow for free expansion.

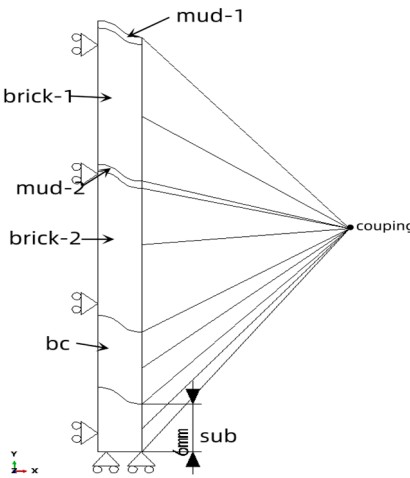

**Figure 4.** Model boundary conditions.

The thermal cycling load on coatings was simulated through convective heat transfer. It is generally accepted that the whole coating remains stress-free when subjected to a high-temperature state that does not cause material phase transition [27,28]. Therefore, before simulating the thermal cycling load, the model was cooled from a high temperature to room temperature to approximate the residual stress generated during the coating preparation process. The specific thermal cycling process was as follows: firstly, the surface of the model was cooled from 1100 °C for 10 min to 25 °C, with a convective heat transfer coefficient of 200 W/(m²·°C). Then, the model surface was heated from 25 °C for 5 min to 1100 °C, with a convective heat transfer coefficient of 500 W/(m²·°C), and the high-temperature state was maintained for 10 min. Thirdly, strong convective cooling was carried out and the model surface was cooled from 1100 °C for 5 min to 25 °C, with a convective heat transfer coefficient of 500 W/(m²·°C).

### 2.2.3. Material Property

The composition and phase constituents of the YSZ "brick" layer and MK resin "mud" layer were detailed in reference [21].

The YSZ "brick" layer is typically characterized as a linear elastic material. The cured "mud" layer is predominantly composed of amorphous $SiO_2$, which is also approximated as a linear elastic material. The BC layer primarily comprises Co, Cr, Al, and Y elements and exhibits elastoplastic behavior, with plastic parameters provided in Table 7 [29].

**Table 7.** Plasticity parameters of BC layer.

| Stress/MPa | Plastic Strain | T/°C |
|---|---|---|
| 1000 | 0.00 | 25 |
| 2500 | 0.230 | 400 |
| 2200 | 0.300 | 600 |
| 375 | 0.022 | 800 |
| 60 | 0.020 | 900 |
| 19 | 0.010 | 1000 |

During the simulation process, isotropic thermal conductivity, thermal expansion, and elasticity were assumed for the materials in each layer. The model assumed a continuous and uniform initial state, disregarding defects resulting from preparation uncertainties. Factors such as creep, high-temperature sintering, and phase transformation stresses were not considered. The material property parameters of each layer in the "brick-mud" layered ceramic-based sealing coating system are shown in Table 8 [21,30].

**Table 8.** Material property parameters of the layers in "brick-mud" layered coating system.

| Category | Substrate | BC Coating | "Brick" Layer | "Mud" Layer |
|---|---|---|---|---|
| Temperature (°C) | 25–1100 | 25–1100 | 25–1100 | 25 |
| Young's modulus (GPa) | 220–120 | 200~110 | 105.5 | 33.4 |
| Poisson's ratio | 0.31~0.35 | 0.30~0.33 | 0.25 | 0.18 |
| Thermal expansion coefficient ($10^{-6}$/K) | 14.8~18.0 | 13.6~17.6 | 9.0~12.2 | 2.2 |
| Thermal conductivity (W/(m·K)) | 88~69 | 5.8~17.0 | 2.0~1.7 | 27 |
| Density (kg/m³) | 8500 | 7380 | 3610 | 2200 |
| Specific heat (J/(kg·K)) | 440 | 450 | 505 | 700 |

### 2.2.4. Extended Finite Element Method

The extended finite element method (XFEM) was utilized to simulate the crack propagation behavior. The Ni-based superalloy SUB layer was assumed to possess a high fracture toughness, rendering it resistant to cracking under operational conditions. Given the predominance of plastic deformation in the BC layer, its cracking behavior was not considered. The "brick" layer and "mud" layer were treated as linear elastic materials, conforming to the traction-separation rule. The maximum principal stress criterion was used to predict crack initiation, expressed as Equation (2).

$$f = \left\{ \frac{<\sigma_{max}>}{\sigma^c_{max}} \right\} \tag{2}$$

where $\sigma_{max}$ is the maximum principal stress. The critical values of the maximum principal stress for the "brick" layer and "mud" layer are 50 MPa and 12 MPa [27], respectively. The $<\sigma_{max}>$ indicates that the element does not undergo damage under pure compressive stress, and its expression is given by Equation (3).

$$\begin{cases} <\sigma_{max}> = 0, \sigma_{max} < 0 \\ <\sigma_{max}> = \sigma_{max}, \sigma_{max} > 0 \end{cases} \tag{3}$$

A power law was employed to assess the mixed-mode behavior of crack growth, which is expressed by Equation (4).

$$\left( \frac{G_n}{G^c_n} \right)^{a_n} + \left( \frac{G_s}{G^c_s} \right)^{a_s} = 1 \tag{4}$$

where $a_n$ and $a_s$ are the coefficients of the power law criterion, and the value of 1 is always assigned to them. $G_n$ and $G_s$ are the strain energy components for pure normal and shear modes, respectively. Assuming that the fracture energy of the material is consistent in all directions, the critical fracture energies for the "brick" layer and "mud" layer were 50 J/m² and 12 J/m², respectively.

## 3. Experimental Results and Discussion

### 3.1. Characterization of "Brick-Mud" Structure Coatings

The cross-sectional microstructure of T1, T2, and T3 coating samples with different thicknesses of "mud" layers is shown in Figure 5. The measurement results of the "mud" layers located on the surface of the TC coatings obtained from Figure 5b,f,l are presented in Table 9. The average thicknesses of the "mud" layers of the T1, T2 and T3 coatings were 2.93 μm, 11.40 μm, and 19.51 μm, respectively. The results indicate that the thickness of "mud" layer on the surface of the TC coatings could be controlled by adjusting the spray volume of the adhesive, as shown in Table 5, and it was proportional to the volume of the adhesive.

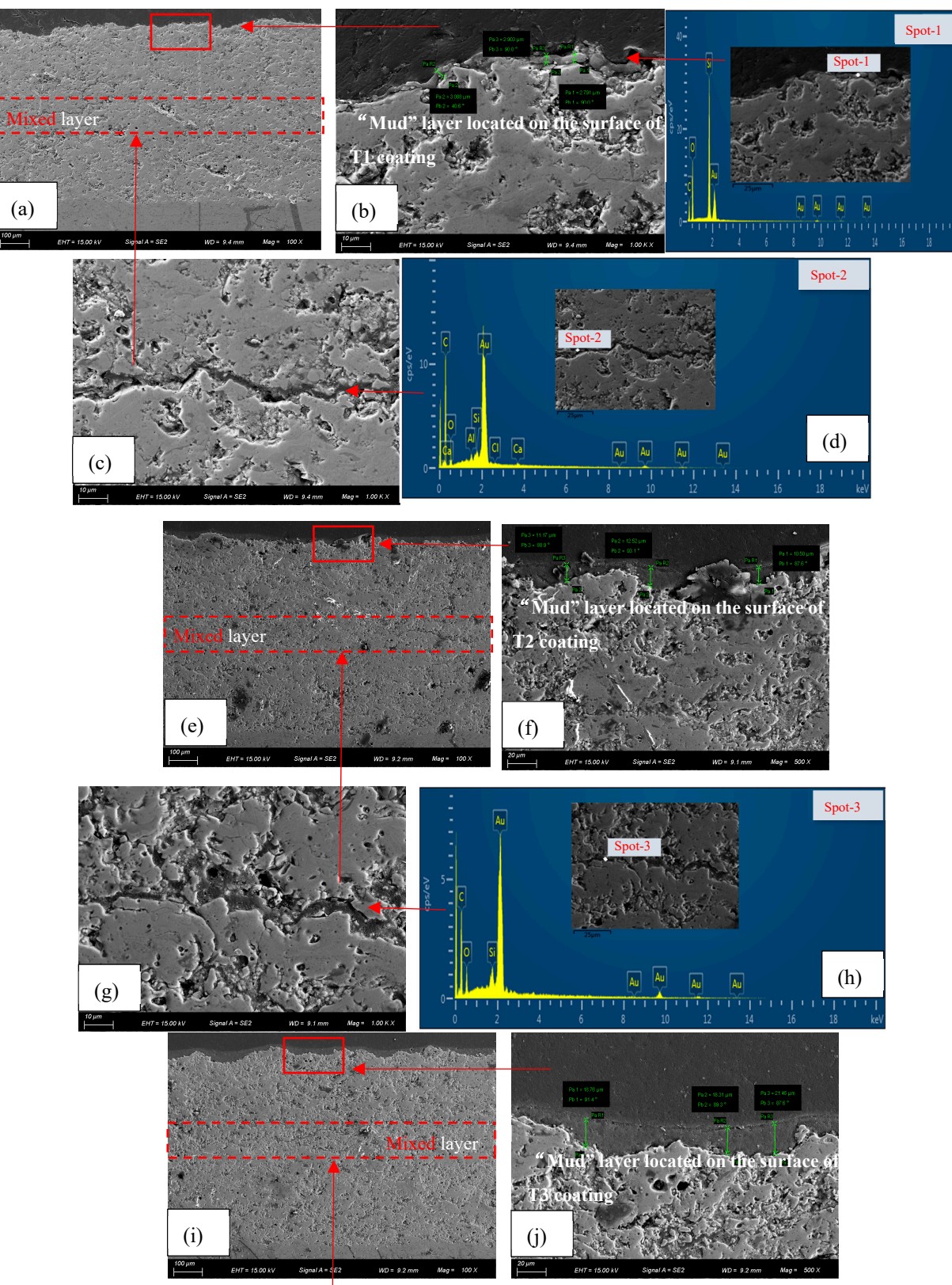

**Figure 5.** *Cont.*

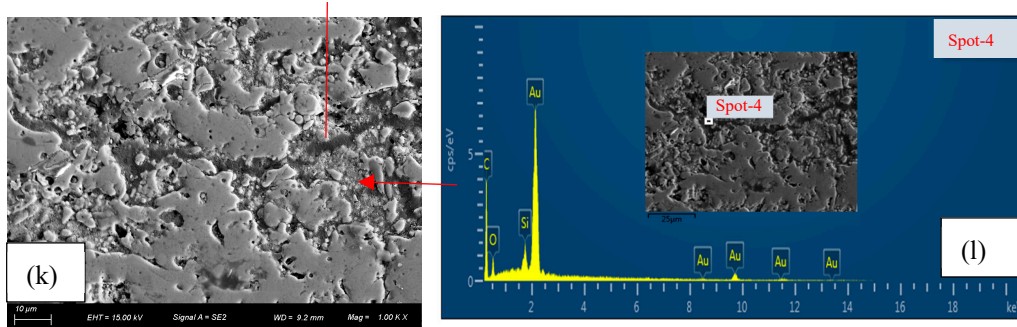

**Figure 5.** Characterization of "mud" layers: (**a**–**d**) characterization of the thin "mud" layer; (**e**–**h**) characterization of the medium-thickness "mud" layer; and (**i**–**l**) characterization of the thick "mud" layer.

When the "mud" layers were located inside the TC coatings, it can be analyzed from Figure 5a,e,i that the three TC coatings prepared all had good physical compatibility (the raw material of the "mud" layer is MK resin, and its characteristic element is Si) between the internal "brick" layer and the "mud" layer with different thicknesses. Especially for T1 coatings with thin "mud" layers, there was almost no interface between the internal "brick" and "mud" layers. For the T2 and T3 coatings with medium and thick "mud" layers, it can be analyzed from Figure 5g,h,k,l that the internal "mud" layers could be partially found inside both the T2 and T3 coatings due to the increase in the "mud" layer thickness.

However, the displayed thicknesses of the internal "mud" layers in the T2 and T3 coatings were significantly smaller than those of the "mud" layers located on the surface of samples, as shown in Figure 5f,j. This was because the thicknesses (Table 9) of the prepared "mud" layers were smaller than the height parameters (Table 6) such as $S_z$, $S_p$, and $S_v$ of the "brick" layers. On the one hand, the "mud" layers would exhibit an undulating and discontinuous state on the rough surface of the "brick" layers. On the other hand, the rough and paste-like "mud" layers would be inevitably impacted by the high-temperature and high-speed particles of the "brick" layer during the subsequent APS process. The molten particles were easy to wrap around the rough surface of the "mud" layer and the unmelted particles were easily to embed in the paste-like "mud" layer. Ultimately, the above factors promoted a good physical compatibility of the internal "brick" and "mud" layers in the TC coatings, and thus formed mixed layers.

**Table 9.** The measurement results of "mud" layers located on the surface of the TC coatings.

| Category | Thickness of the "Mud" Layer in Three Measurements/μm | | | Average Thickness/μm |
|---|---|---|---|---|
| T1 | 3.088 | 2.903 | 2.791 | 2.93 |
| T2 | 11.17 | 12.52 | 10.50 | 11.40 |
| T3 | 18.76 | 18.31 | 21.46 | 19.51 |

### 3.2. Bonding Performance of "Brick-Mud" Structure Coatings

Bond strength is an important performance parameter for sealing coating. The "mud" layer exhibited a lower fracture toughness and served as the "weaker" layer in the "brick-mud" layered ceramic-based sealing coatings. When the "mud" layer thickness was increased, it would potentially decrease the bonding strength of the coating. Therefore, this study conducted tensile tests on the T1, T2, and T3 coatings to investigate the effect of the "mud" layer thickness on their failure modes and bonding performance.

The typical tensile fracture patterns and EDS results of a conventional structure coating (without "mud" layers), T1 coating, T2 coating, and T3 coating are shown in Figure 6. It can be analyzed from Figure 6a that the fracture of the conventional structure coating mainly consisted of a central area and edge area, and the central area was much larger than the

edge area. The EDS test results in Figure 6a indicate that the central area mainly contained elements such as Zr, Al, O, Co, and Cr, etc.

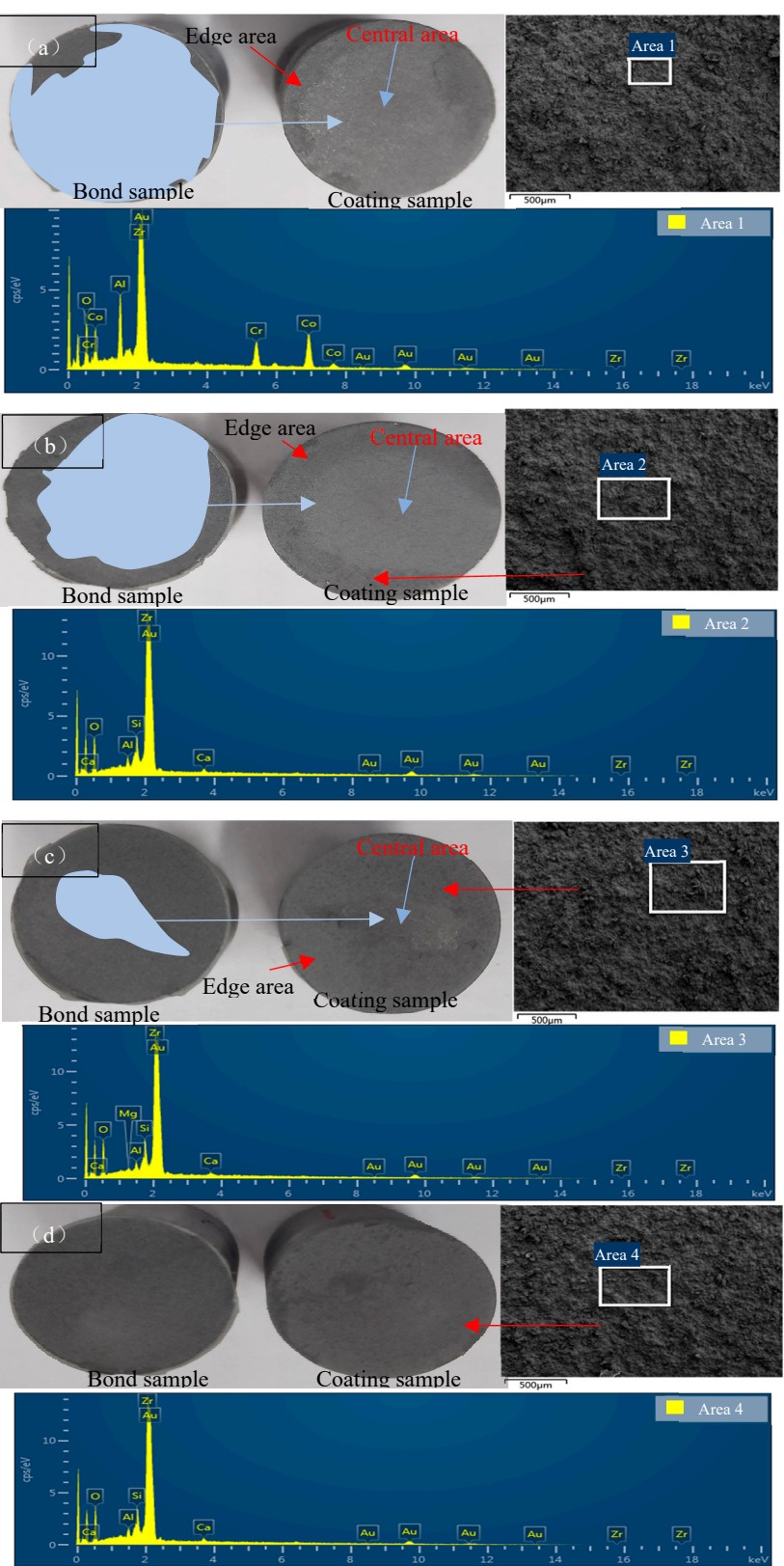

**Figure 6.** Typical tensile fracture patterns and EDS results of the bond strength samples: (**a**) conventional coating; (**b**) T1 coating; (**c**) T2 coating; and (**d**) T3 coating.

The above results demonstrate that the fracture location of the conventional structural coating was mainly located at the TC/BC interface. During the tensile process, cracks first originated in the central region of the coating, and then gradually expanded from the center to the surrounding area with the increase in the tensile force, ultimately breaking off instantly at the edge of the coating. The tensile fracture characteristics of the T1 specimens were similar to the conventional structural samples, which were mainly composed of a central area and edge area. The difference was that the T1 fracture had a smaller central area and larger edge area. The EDS results in Figure 6b indicate that the edge area mainly contained elements such as Zr, O, and Si, etc. The above results prove that the edge area of the T1 specimens was located at the "brick" layer/"mud" layer interface. When the thickness of the "mud" layer increased to about 10 μm, it could be analyzed from Figure 6c that only a small amount of breaking occurred in the central area of the T2 fracture, and its fracture location was mainly at the interface between the "brick" layer and "mud" layer. As the thickness of the "mud" layer further increased, it could be analyzed from Figure 6d that the fracture position of the T3 specimen was almost entirely located at the "brick" layer/"mud" layer interface.

Based on the above analysis, the tensile fracture mode of T1 coatings was not much different from that of conventional structure coatings. However, as the thickness of "mud" layer increased, the fracture positions of tensile samples would gradually be transferred from the TC/BC interface to the "brick" layer/"mud" layer interface inside the "brick-mud" layered structure TC coating.

The load–displacement curves of the T1, T2, T3, and conventional structure coatings are shown in Figure 7. The bond strengths of the four types of coatings were 7.13 MPa, 6.27 MPa, 5.32 MPa, and 3.66 MPa, on average, respectively.

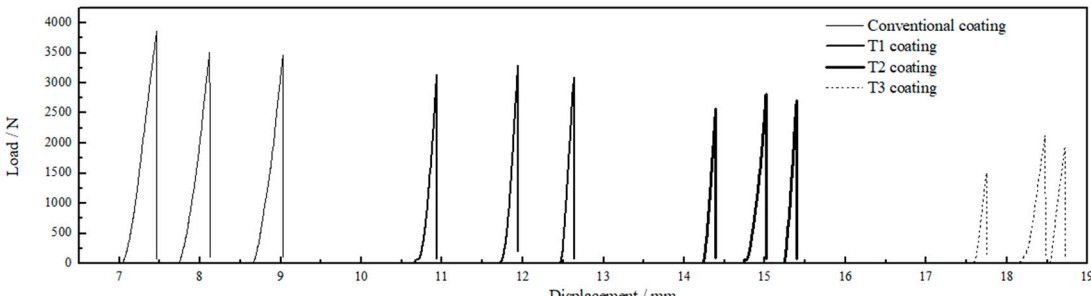

**Figure 7.** Load–displacement curves of T1, T2, T3, and conventional coatings.

The bonding strength of the ceramic-based sealing coatings gradually decreased with the increase in the "mud" layer thickness. Notably, when the "mud" layer thickness reached about 20 μm, the bonding strength of the T3 coating exhibited a significant decrease of 48.67% compared with the conventional coating.

### 3.3. Thermal Cycling Performance of "Brick-Mud" Structure Coatings

Figure 8 shows macroscopic images of the T0 (conventional structure coating), T1, T2, and T3 coatings after a failure of thermal cycling. Table 10 presents the thermal cycling test results of the above four types of coatings. The T0 coating demonstrated a thermal cycling life of only 70 cycles with complete delamination along the TC/BC interface as the failure mode. The T1 and T2 coatings had the same failure mode as the T0 coating, but their thermal cycling lives increased by 90.0% and 135.7%, respectively. However, the thermal cycling life of the T3 coating decreased by 81.4% compared with the T0 coating, and the failure mode changed from overall spalling of the TC coating to spalling of the first "brick" layer.

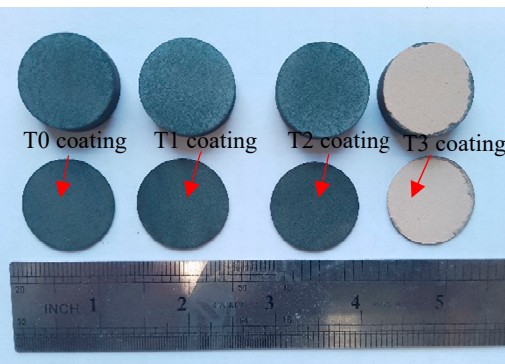

**Figure 8.** Macroscopic images of the four types of coatings after thermal cycling failure.

**Table 10.** Thermal cycling test results of coatings.

| Samples | Number of Thermal Cycles | | | | Failure Modes |
|---|---|---|---|---|---|
| | Specimen 1 | Specimen 2 | Specimen 3 | Average Values | |
| Conventional ceramic-based sealing coating (T0) | 77 | 72 | 62 | 70 | Spalling of the ceramic coating along TC/BC interface |
| T1 | 141 | 127 | 130 | 133 | Spalling of the ceramic coating along TC/BC interface |
| T2 | 158 | 176 | 160 | 165 | Spalling of the ceramic coating along TC/BC interface |
| T3 | 12 | 12 | 16 | 13 | Spalling of the first "brick" layer |

The cross-sectional microstructures of the T0, T1, T2, and T3 coatings after different thermal cycles are shown in Figure 9. Figure 9a shows the cross-sectional microstructure of the T0 coating after 60 thermal cycles. There were very few microcracks generated in the T0 coating with a continuous stacked structure, while a big and continuous transverse crack appeared at the TC/BC interface, which eventually led to the overall spalling of the conventional structure TC coating. Figure 9b shows the cross-sectional microstructure of T3 coating. After only 10 thermal cycles, a large transverse crack appeared in the thick "mud" layer inside the TC coating, ultimately causing the first "brick" layer in the T3 coating to peel off and fail.

The cross-sectional microstructures of the T1 coating are shown in Figure 9c,d, respectively. Multiple longitudinal microcracks appeared in the TC coating after 60 thermal cycles (no obvious transverse cracks between layers). The TGO growth stress was relatively low (due to fewer thermal cycles and shorter thermal oxidation hours) at this stage, resulting in no significant cracking between the TC coating and BC coating. When the number of thermal cycles was increased to 120, on the one hand, the sizes of the longitudinal cracks in the T1 coating were significantly increased, and there were still no obvious transverse cracks. On the other hand, the TGO growth stress gradually increased due to the continuous prolongation of thermal oxidation hours with the increase in thermal cycles, resulting in obvious large-sized transverse cracks appearing between the TC/BC interface.

The cross-sectional microstructures of the T2 coating are shown in Figure 9e,f, respectively. When the number of thermal cycles was 60, multiple longitudinal microcracks and smaller transverse cracks appeared in the T2 coating. There was also no significant cracking in the TC/BC interface of the T2 coating at this stage. When the number was increased to 120, on the one hand, in addition to a significant increase in the size of the longitudinal cracks, evident transverse cracking also occurred at the "mud" layer inside the T2 coating. On the other hand, obvious large transverse cracks also appeared between the TC coating and BC coating.

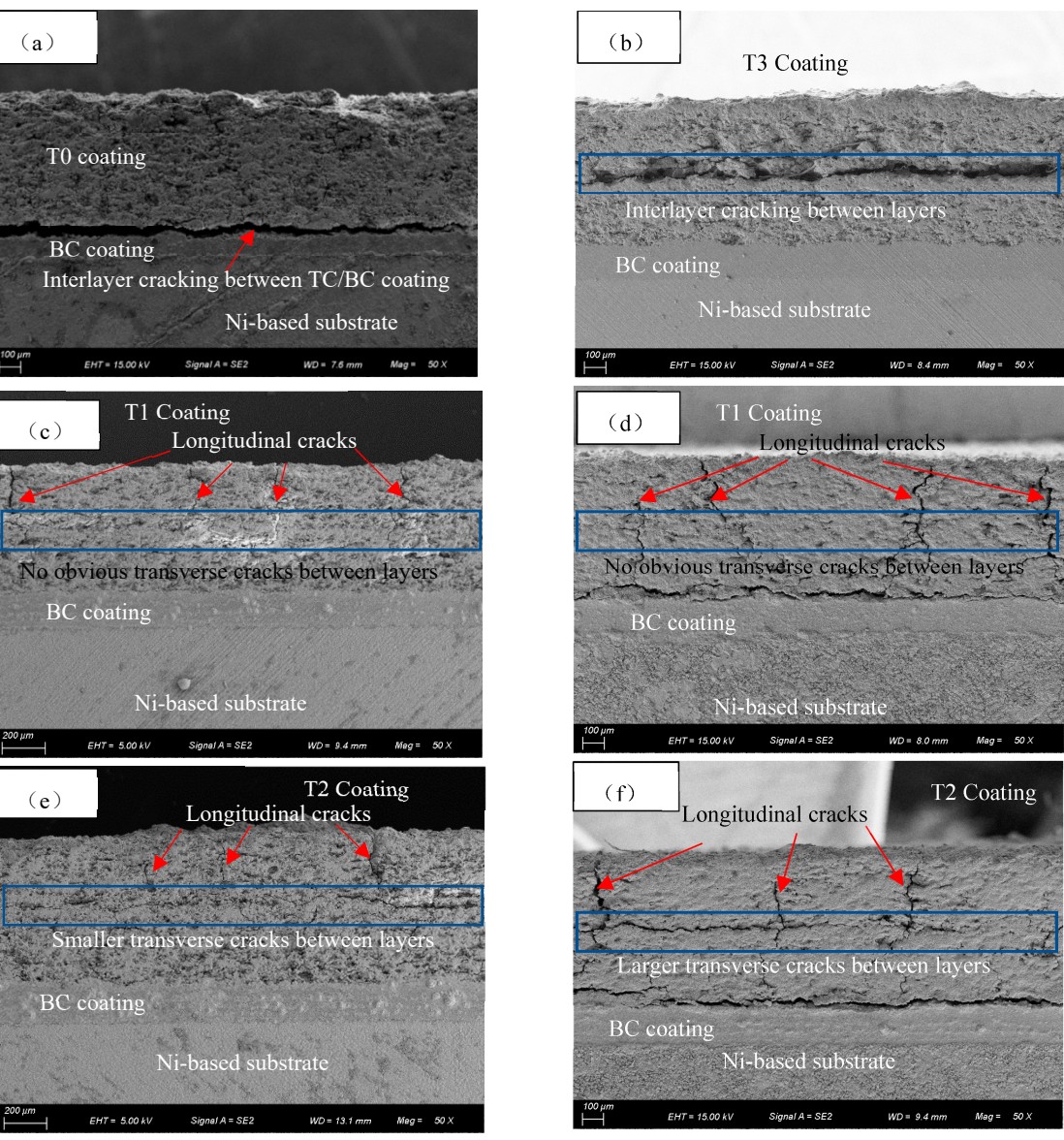

**Figure 9.** Propagation behavior of thermal cycling cracks in the four types of coatings: (**a**) 60 cycles for T0 coating; (**b**) 10 cycles for T3 coating; (**c**) 60 cycles for T1 coating; (**d**) 120 cycles for T1 coating; (**e**) 60 cycles for T2 coating; and (**f**) 120 cycles for T2 coating.

Due to the low fracture toughness, the uncovered "mud" layer was prone to falling off after thermal cycling, which was not conducive to analyzing the initiation behavior of the cracks or proving that the "mud" layer was the source of the cracks. Therefore, the T3 coating with a thick "mud" layer before thermal cycling was selected for SEM and EDS testing, and the results are shown in Figure 10. It can be analyzed from Figure 10a that some micro cracks occurred in the T3 coating due to thermal spraying stress and cutting stress. Figure 10b shows the elemental mapping scanning result of the cracking area. It can be seen that the purple area was the uncovered "mud" layer. The above test results demonstrate that the "mud" layer inside the TC coating with a "brick- mud" layered structure has the function of micro cracking, and would preferentially initiate micro cracks under stress.

In summary, the addition of "mud" layers with a low fracture toughness led to the initiation and propagation of longitudinal cracks in the TC coating during thermal cycling, which could consume part of the internal stress during the thermal cycling process and reduce the stress concentration at the BC/TC interface, thereby improving the thermal cycling performances of the T1 and T2 "brick-mud" structure coatings. However, as the

thickness of the "mud" layer increased, the trend of thermal cycling lateral cracking in the "mud" layer inside the TC coating became stronger. Especially when the thickness of the "mud" layer reached about 20 μm, the thermal cycling performance of the T3 coating was sharply decreased due to the initiation and rapid propagation of transverse cracks.

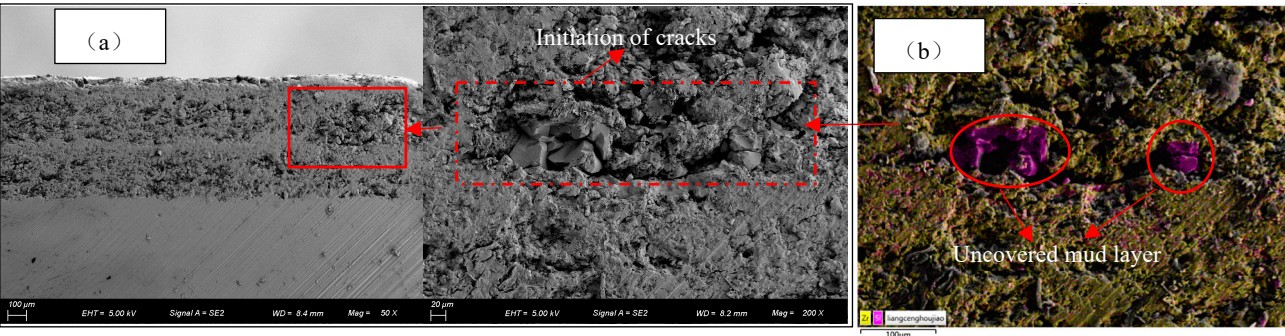

**Figure 10.** The initiation of cracks in the "brick-mud" structure coating: (**a**) initiation of cracks; and (**b**) elemental mapping scanning result.

## 4. Simulation Calculation Results and Analysis

### 4.1. Distribution and Evolution Rules of Thermal Cycling Stress

The stress field during the thermal cycling process is the driving force that leads to the initiation and propagation of cracks. Therefore, investigating the stress field of "mud" layers with different thicknesses in TC coatings with "brick-mud" structures is of great significance for analyzing and predicting the initiation and propagation rules of thermal cycling cracks in a new structure coating. Because pores and "mud" layers are the bivariate that can affect the stress field of the model, this paper first studied the distribution and evolution of thermal cycling stress field in a dense structure model (no pores in the "brick" layer, and the "mud" layer was added as a single variable). Then, pores were added to establish a model of the "brick" layer with a porous structure, and the stress field distribution and evolution rules of the porous structure model were analyzed. It should be pointed out that, as a sacrificial coating, the cracking and spalling of the M1 layer located on the surface of the coating would not have an adverse impact on the function of the sealing coating. Therefore, this paper mainly focused on the M2 layer located between the B1 and B2 layers.

#### 4.1.1. Stress Distribution of Normal Stress $\sigma_{22}$ and Shear Stress $\sigma_{12}$

(1)  Distribution of $\sigma_{22}$ and $\sigma_{12}$ in the dense model

The initiation and continuous propagation of transverse cracks inside TC coatings are the main reasons for the interlayer delamination failure of the coating. The driving force for the initiation and propagation of transverse cracks during thermal cycling mainly includes normal tensile stress $\sigma_{22}$  and shear stress $\sigma_{12}$.

The distribution nephograms of $\sigma_{22}$ after thermal cycling in the dense models are shown in Figure 11. The $\sigma_{22\ max}$ of the M2$_t$ (the thin M2 layer located between the B1 and B2 layers), M2$_M$ (the medium-thickness M2 layer located between the B1 and B2 layers), and M2$_T$ (the thick M2 layer located between the B1 and B2 layers) "mud" layers was 9.68 MPa, 18.16 MPa, and 36.79 MPa, respectively. The tensile stress borne by the M2 layer gradually increased as the thickness of the M2 layer increased. The maximum tensile stress of the M2$_t$ layer was mainly concentrated in the peaks and valleys. As the thickness of the M2 layer increased, the concentration area gradually transferred towards the interface.

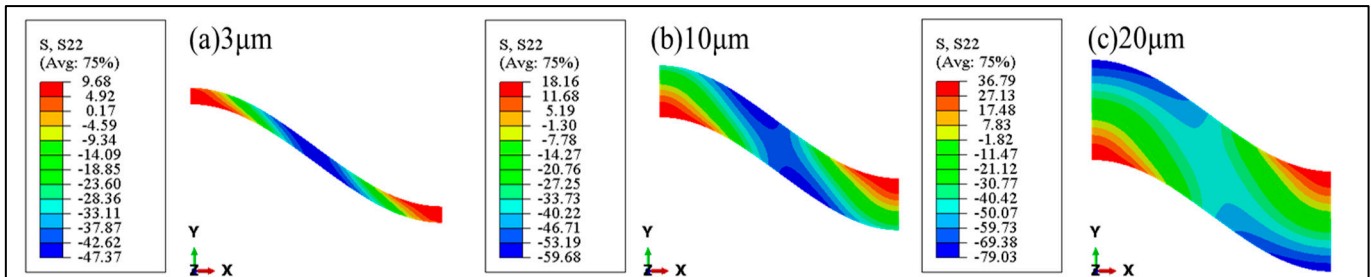

**Figure 11.** Distribution of $\sigma_{22}$ in dense models: (**a**) M2$_t$; (**b**) M2$_M$; and (**c**) M2$_T$.

Shear stress $\sigma_{12}$ is the dominant stress for the initiation and propagation of type II interface cracks (sliding cracks). The distribution nephograms of $\sigma_{12}$ after thermal cycling in the dense models are shown in Figure 12. The $\sigma_{12\ max}$ of the M2$_t$, M2$_M$, and M2$_T$ "mud" layers was 60.63 MPa, 82.85 MPa, and 94.90 MPa, respectively. As the thickness of the M2 layer increased, the shear stress borne by the M2 layer also gradually increased. The shear stress of the M2$_t$ layer was mainly concentrated in the middle region between peaks and valleys, while the concentration area gradually transferred towards the interface as the thickness of the M2 layer increased.

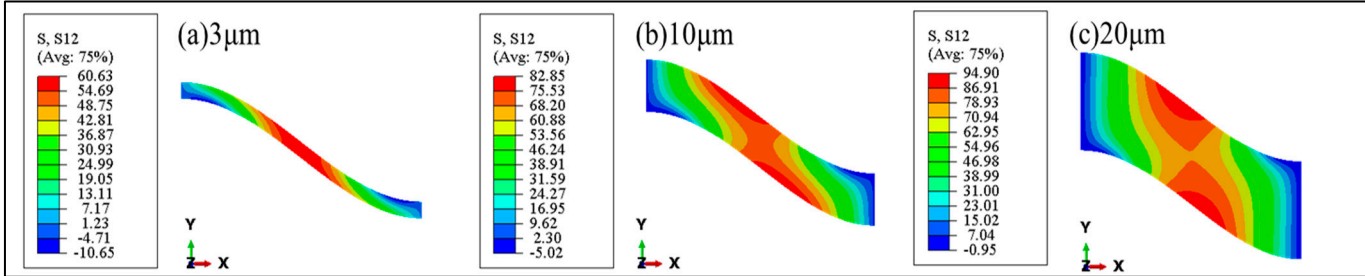

**Figure 12.** Distribution of $\sigma_{12}$ in dense models: (**a**) M2$_t$; (**b**) M2$_M$; and (**c**) M2$_T$.

(2)     Distribution of $\sigma_{22}$ and $\sigma_{12}$ in the porous model

The distribution nephograms of $\sigma_{22}$ after thermal cycling in the porous models are shown in Figure 13. The $\sigma_{22\ max}$ of the M2$_t$, M2$_M$, and M2$_T$ "mud" layers was 157.29 MPa, 78.42 MPa, and 95.22 MPa, respectively. As the thickness of the M2 layer increased, the tensile stress borne by the M2 layer first decreased and then increased. The tensile stress of the M2$_t$ layer was mainly concentrated in the middle region between peaks and valleys. As the thickness of the M2 layer increased, on the one hand, the concentration area of tensile stress gradually transferred towards the interface. On the other hand, there were multiple concentration locations of tensile stress in both the M2$_M$ and M2$_T$ "mud" layers.

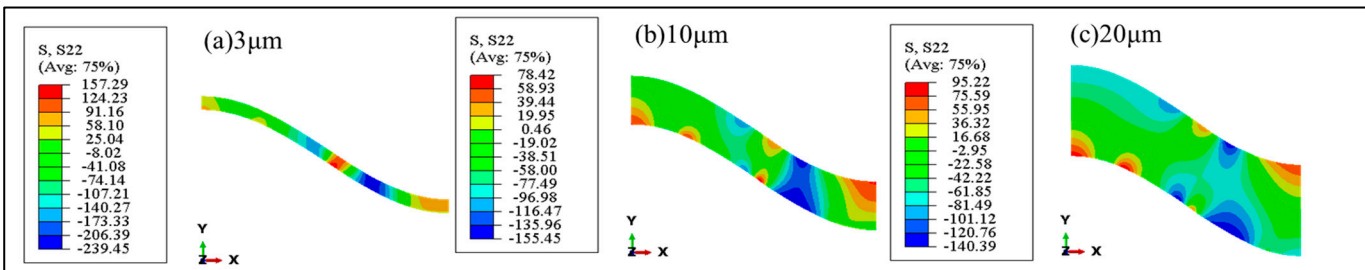

**Figure 13.** Distribution of $\sigma_{22}$ in porous models: (**a**) M2$_t$; (**b**) M2$_M$; and (**c**) M2$_T$.

The distribution nephograms of $\sigma_{12}$ after thermal cycling in the porous models are shown in Figure 14. The $\sigma_{12\ max}$ of the M2$_t$, M2$_M$, and M2$_T$ "mud" layers was 250.27 MPa,

208.77 MPa, and 210.26 MPa, respectively. As the thickness of the M2 layer increased, the shear stress borne by the M2 layer first rapidly decreased and then slowly increased. The shear stress of the $M2_t$ layer was mainly concentrated in the middle region between peaks and valleys, while the concentration area also gradually transferred towards the interface as the thickness of the M2 layer increased.

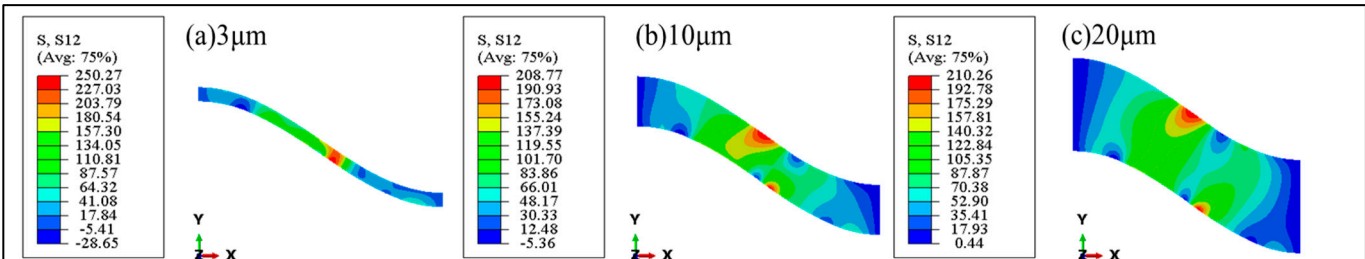

**Figure 14.** Distribution of $\sigma_{12}$ in porous models: (**a**) $M2_t$; (**b**) $M2_M$; and (**c**) $M2_T$.

4.1.2. Effect of the "Mud" Thickness on $\sigma_{22}$ and $\sigma_{12}$

(1)    The changing trend of stress concentration location of $\sigma_{22}$ and $\sigma_{12}$

For both dense and porous models, the stress concentration positions of the tensile and shear stresses borne by the M2 layer transferred towards the interface with the increase in the thickness of the "mud" layer. The difference was that the number of stress concentration positions in the dense model remained unchanged as the thickness of the "mud" layer increased, while in the porous model, there were multiple locations of tensile stress concentration in both the $M2_M$ and $M2_T$ "mud" layers due to the effect of pores in the "brick" layer.

(2)    The stress evolution rules of $\sigma_{22\ max}$ and $\sigma_{12\ max}$

Stress concentration is the main reason for crack initiation and propagation; therefore, this paper mainly analyzed the evolution rules of the maximum stress. The comparison bar charts of $\sigma_{22\ max}$ and $\sigma_{12\ max}$ borne by the M2 layers with different thicknesses in dense and porous models are shown in Figure 15.

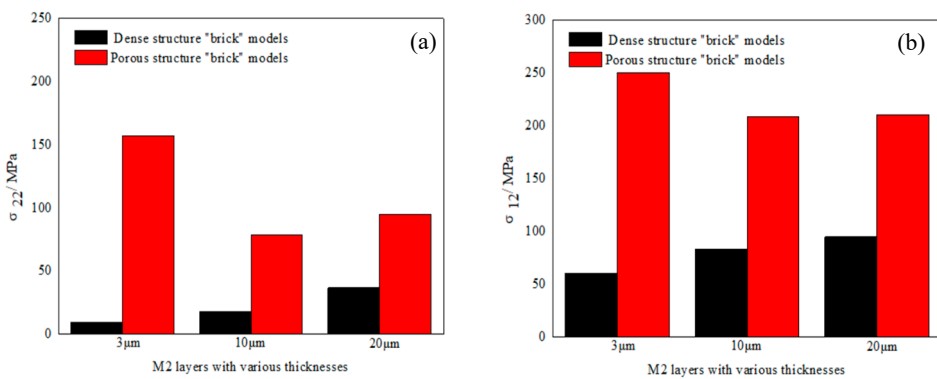

**Figure 15.** Comparison bar charts in dense and porous models: (**a**) $\sigma_{22}$; and (**b**) $\sigma_{12}$.

When the "brick" layer model was a dense structure, the difference in the thermal expansion coefficient between the dense "brick" layer and the "mud" layer was the main factor that generated $\sigma_{22}$ and $\sigma_{12}$. Because the degree of expansion or contraction of the "mud" layer during thermal cycling increased with the increase in the "mud" layer thickness, the $\sigma_{22\ max}$ and $\sigma_{12\ max}$ borne by the M2 layer in the dense model also gradually increased.

When the "brick" layer model was a porous structure, on the one hand, the $\sigma_{22\ max}$ and $\sigma_{12\ max}$ both significantly increased compared with the dense model. On the other hand, the stress evolution rules were different from the dense models. As the thickness of

the "mud" layer increased, the $\sigma_{22\ max}$ showed a trend of first rapidly decreasing and then slowly increasing, while the $\sigma_{12\ max}$ showed a trend of first decreasing and then remaining basically unchanged. The above results indicate that the addition of pores in the "brick" layer changed the $\sigma_{22}$ and $\sigma_{12}$ stress fields of the M2 layer in the porous models. In addition to the difference in the thermal expansion coefficient between the porous "brick" layer and the "mud" layer, the porous in the "brick" layer was the second main factor affecting the $\sigma_{22}$ and $\sigma_{12}$ stress fields.

4.1.3. Distribution of Normal Stress $\sigma_{11}$

The longitudinal cracks inside coatings are mainly subjected to normal tensile stress $\sigma_{11}$ parallel to the coating interface. Longitudinal cracks are similar to columnar crystal gaps in electron beam physical vapor deposition (EB-PVD) ceramic coatings, which have the function of releasing internal stress during thermal cycling and can improve the strain tolerance and thermal cycling life of the coating [31,32]. Therefore, research on $\sigma_{11}$ is of great significance for analyzing and predicting the initiation and propagation rules of longitudinal cracks in a new structure coating.

Figure 16 shows $\sigma_{11}$ stress nephograms in the dense and porous models after the thermal cycling process. Both the dense model and the porous model exhibited overall normal compressive stress. For the porous model, $\sigma_{11}$ was observed only near the pores, the influence range of which was extremely small. This was because the thermal expansion coefficients of the SUB and BC layers were much greater than those of the "brick" and "mud" layers, and the interior of the coating was subjected to normal compressive stress from the SUB and BC layers during the cooling stage. In the instantaneous cooling stage (defined as about 500 ms after the start of cooling), there was a significant temperature difference between the surface and the interior of the model. The huge temperature difference may cause insufficient shrinkage of the SUB and BC layers, thereby affecting the $\sigma_{11}$ stress field in the model. Therefore, this paper mainly researched the distribution rules of $\sigma_{11}$ after the instantaneous cooling stage.

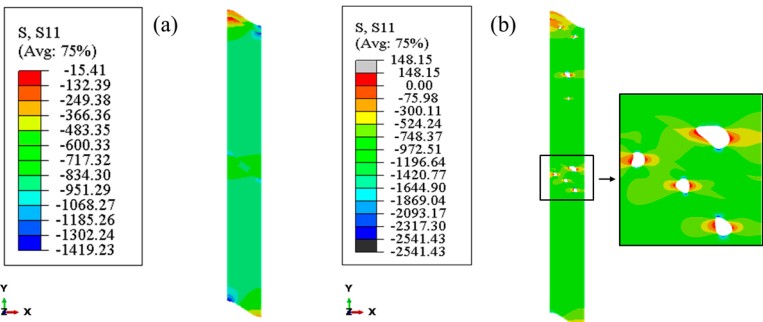

**Figure 16.** Distribution of $\sigma_{11}$: (**a**) dense model; and (**b**) porous model.

(1)  Distribution of $\sigma_{11}$ in the dense model

The distribution nephograms of $\sigma_{11}$ in the dense models after the instantaneous cooling stage are shown in Figure 17. The maximum $\sigma_{11}$ of the M2$_t$, M2$_M$, and M2$_T$ "mud" layers was 14.14 MPa, 6.77 MPa, and 3.88 MPa, respectively.

As the thickness of the M2 layer increased, the $\sigma_{11}$ borne by the M2 layer gradually decreased. The $\sigma_{11}$ of the M2$_t$ layer was mainly concentrated in the middle region between peaks and valleys, while the concentration area gradually transferred towards the upper surface of the peak and the undersurface of the valley as the thickness of the M2 layer increased.

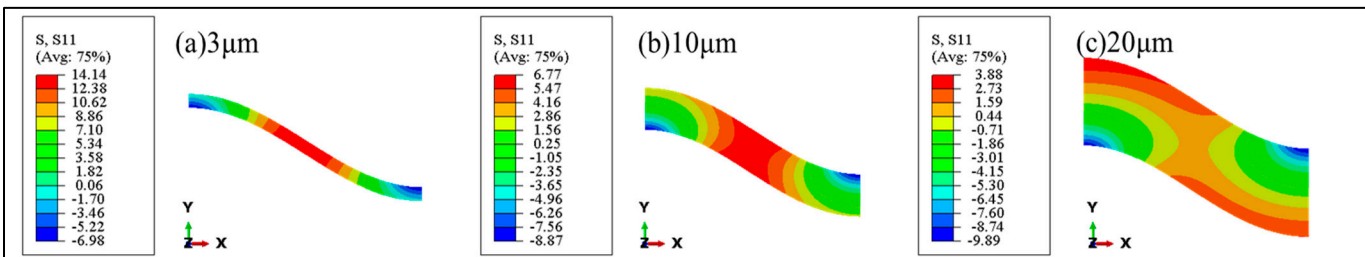

**Figure 17.** Distribution of $\sigma_{11}$ after the instantaneous cooling stage in the dense models: (**a**) M2$_t$; (**b**) M2$_M$; and (**c**) M2$_T$.

(2)  Distribution of $\sigma_{11}$ in the porous model

The distribution nephograms of $\sigma_{11}$ in the porous models after instantaneous cooling stage are shown in Figure 18. The maximum $\sigma_{11}$ of the M2$_t$, M2$_M$, and M2$_T$ "mud" layers was 13.83 MPa, 10.70 MPa, and 5.43 MPa, respectively. As the thickness of the M2 layer increased, the $\sigma_{11}$ borne by the M2 layer also gradually decreased. The $\sigma_{11}$ of the M2$_t$ layer was mainly concentrated in the middle region between peaks and valleys. As the thickness of the M2 layer increased, the concentration of normal tensile stress gradually transferred towards the interface. In addition, there was a trend of $\sigma_{11}$ concentration at the interface near the peak and valley of the M2$_T$ layer.

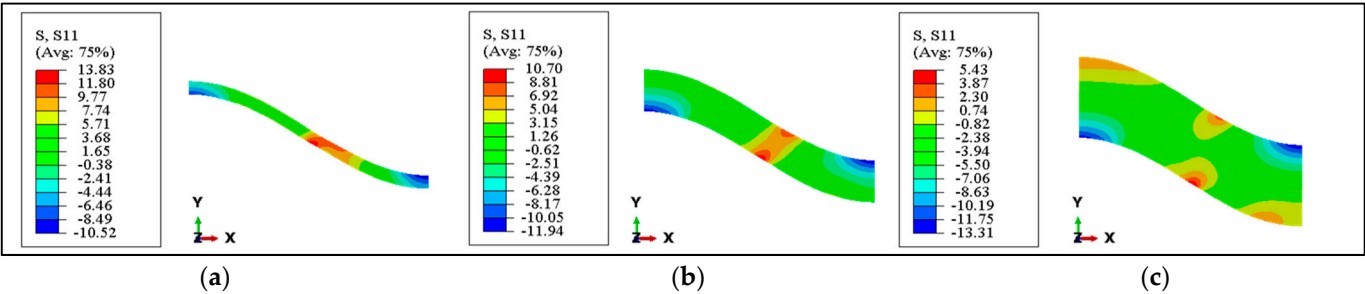

**Figure 18.** Distribution of $\sigma_{11}$ after the instantaneous cooling stage in the porous models: (**a**) M2$_t$; (**b**) M2$_M$; and (**c**) M2$_T$.

4.1.4. Effect of the "Mud" Thickness on $\sigma_{11}$

(1)  The changing trend of stress concentration location of $\sigma_{11}$

For both dense and porous models, the stress concentration positions of $\sigma_{11}$ borne by the M2 layer transferred towards the interface with the increase in the thickness of "mud" layer. The difference was that the number of stress concentration positions in the dense model remained unchanged as the thickness of the "mud" layer increased, while in the porous model, there were multiple locations of normal stress concentration in the M2$_T$ layer due to the effect of pores in the "brick" layer.

(2)  The stress evolution rules of $\sigma_{11\ max}$

The comparison bar charts of the $\sigma_{11\ max}$ borne by the M2 layers with different thicknesses in dense and porous models are shown in Figure 19. As the thickness of the "mud" layer increased, the $\sigma_{11\ max}$ borne by the M2 layer in both models gradually decreased. The difference was that the reduction rate of $\sigma_{11\ max}$ in the dense models was greater than that in the porous models. The $\sigma_{11\ max}$ of the M2$_t$ layer in the dense and porous models were basically equivalent, while the $\sigma_{11\ max}$ of the M2$_M$ and M2$_T$ layers in the porous models were significantly higher than that in the dense models. The above results indicate that the addition of pores in the "brick" layer also changed the $\sigma_{11}$ stress field of the M2 layer. In addition to the difference in the thermal expansion coefficient between the porous "brick" layer and the "mud" layer, the porous structure was also the second main factor affecting the $\sigma_{11}$ stress field.

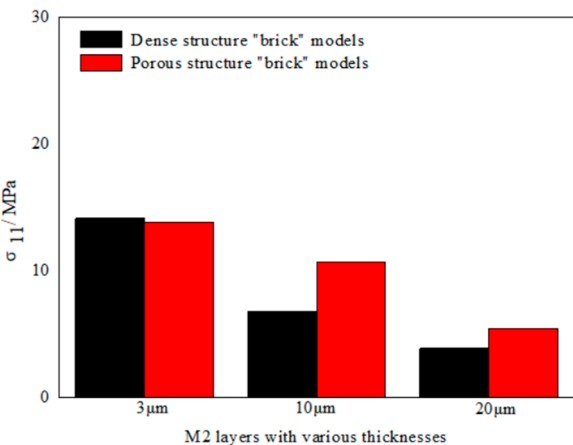

**Figure 19.** $\sigma_{11}$ comparison bar charts in dense and porous models.

### 4.2. *The Effect of M2 Layer on Crack Propagation Behavior in Porous Models*

4.2.1. Evolution Rules of Longitudinal Cracks Propagation

The prefabricated longitudinal crack in the porous model with the $M2_t$ layer is shown in Figure 20a. After undergoing a complete thermal cycle, the crack propagation paths in the T0 (without a "mud" layer), T1 (with the $M2_t$ layer), T2 (with the $M2_M$ layer), and T3 (with the $M2_T$ layer) models are shown in Figure 20b–e, respectively.

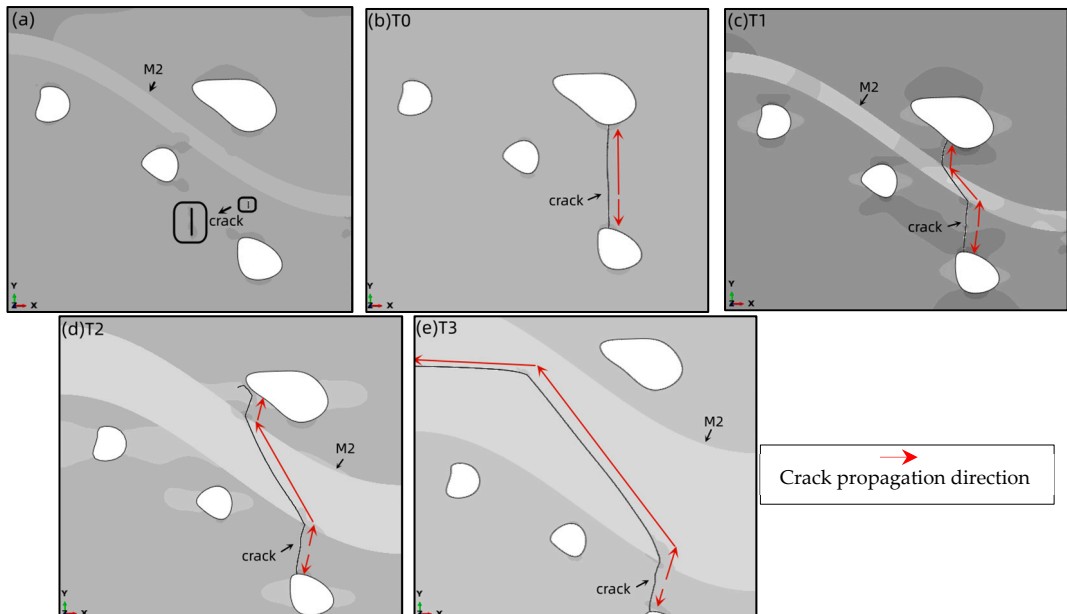

**Figure 20.** Prefabricated longitudinal crack and crack propagation paths in models: (**a**) prefabricated longitudinal crack; (**b**) T0 model; (**c**) T1 model; (**d**) T2 model; and (**e**) T3 model.

It can be analyzed from Figure 20b that the longitudinal crack propagated in a straight line along the y-direction during thermal cycling and eventually terminated at the pores in the T0 model. For the T1, T2, and T3 models, when the longitudinal crack propagation paths encountered different thicknesses of "mud" layers, all of the propagation directions were changed from a longitudinal direction to an approximately 45° angle with the y-axis. After passing through the $M2_t$ layer, the longitudinal crack continued to propagate longitudinally and terminate at the pores in the T1 model. In the T2 model, the longitudinal crack continued to propagate longitudinally after passing through the $M2_M$ layer and underwent a secondary deflection at the edge of the pores, while in the T3 model, a

secondary deflection occurred along the negative *x*-axis direction inside the M2$_T$ layer due to the large thickness of the "mud" layer.

Figure 21 shows the comparison bar charts of the thermal cycling energy dissipation of models with different thicknesses of "mud" layers. Under the unchanged conditions of the location of the prefabricated crack and the distance between pores and the "mud" layer, the energy dissipation of the T0 model was $1.37 \times 10^{-5}$ mJ, while that of T1, T2, and T3 models was increased by 14.1%, 428.1%, and 1096.7%, respectively, compared with the T0 model.

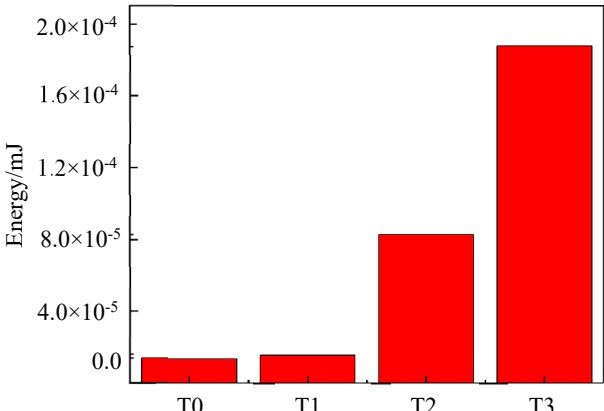

**Figure 21.** Energy dissipation levels of models with different thicknesses of "mud" layer.

### 4.2.2. Evolution Rules of Transverse Crack Propagation

When a transverse crack appears in the "mud" layer, its lower fracture toughness may cause the continuous growth of transverse cracks in the M2 layer, leading to interlayer delamination failure of the "brick-mud" layered structure coating. In order to study the effect of "mud" layer thickness on the transverse crack propagation behavior of the M2 layer, a transverse crack with a length of 1 μm was preset in the porous model, as shown in Figure 22a.

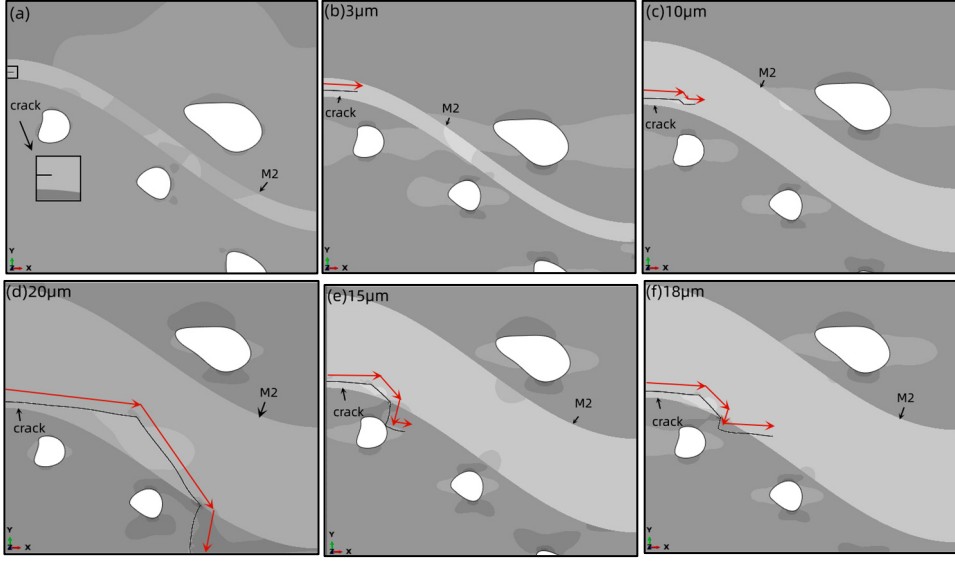

→ Crack propagation direction

**Figure 22.** Prefabricated transverse crack and crack propagation paths in models: (**a**) prefabricated transverse crack; (**b**) T1 model; (**c**) T2 model; (**d**) T3 model; (**e**) model with 15 μm thickness of "mud" layer; and (**f**) model with 18 μm thickness of "mud" layer.

After undergoing a thermal cycling process, it can be analyzed from Figure 22b–d that the transverse crack in the T1 model propagated in a straight line along the *x*-axis direction, and the crack length was the shortest. After extending a certain length along the *x*-axis direction, the transverse crack in the T2 model underwent a small deflection at an angle of about 45° to the negative direction of the *y*-axis, then continued to propagate laterally in the "mud" layer, and the crack length was still relatively short. After extending for a longer length along the *x*-axis direction, the transverse crack in the T3 model underwent a significant deflection at a 45° angle to the negative *y*-axis direction until it reached the B2/M2 interface, and then continued to propagate along the negative *y*-axis direction. The above results demonstrate that the crack length in the T3 model was significantly increased compared with the T1 and T2 models.

In order to accurately determine the critical "mud" layer thickness for the significant propagation of transverse cracks, the growth paths of transverse cracks in models with a 15 µm thickness and 18 µm thickness of the "mud" layer are shown in Figure 22e,f. The research results indicate that, when the thickness was less than 18 µm, the propagation length of the transverse crack slowly increased with the increase in the thickness of the "mud" layer. However, when the thickness increased to 20 µm, the propagation length of the transverse crack increased markedly.

*4.3. The Failure Mechanism of Transverse Cracking in the Thick "Mud" Layer*

Based on the above experimental and simulation research results, the initiation and propagation behavior of cracks in the "brick-mud" layered structure TC coating was mainly affected by the stress field and the interface height parameters. Among them, the stress field was the driving force for crack initiation and propagation, and the interface height parameters affected the propagation path of cracks. The analysis of the influence of the stress field on the crack propagation behavior in different thicknesses of "mud" layers was as follows. Based on the stress calculation results in Figure 15, the $M2_t$ layer bore the largest $\sigma_{22\ max}$ and $\sigma_{12\ max}$ (the two main driving forces that promote transverse crack growth), while the $M2_T$ layer bore smaller and the $M2_M$ layer bore the smallest of the two stresses. Based on the stress calculation results in Figure 19, the $M2_t$ layer bore the largest $\sigma_{11\ max}$ (the main driving force for longitudinal crack growth), while the $M2_M$ layer bore smaller and the $M2_T$ layer bore the smallest of the stress. The above stress calculation results imply that the initiation trend of transverse cracks was ranked in descending order from the thin "mud" layer, thick "mud" layer, to the medium thickness "mud" layer, while the initiation trend of longitudinal cracks was ranked in descending order from the thin "mud" layer, medium thickness "mud" layer, to the thick "mud" layer.

The analysis of the influence of interface height parameters on the transverse crack propagation behavior in different thicknesses of "mud" layers was as follows. Due to the fluidity of MK resin before solidification, when the thickness of the "mud" layer was thin, a discontinuous "mud" layer on the surface of the rough "brick" layer was formed, as shown in Figure 23a. The sediment layer near the trough was thicker, while that near the peak was thinner. When the transverse crack in the thin "mud" layer propagated near the peak, it was suppressed by the "brick" layer with a higher fracture toughness, thereby changing the direction of crack propagation or terminating it. The schematic diagram of the formation of a thick "mud" layer on the surface of the "brick" layer is shown in Figure 23b. When the thickness of the "mud" layer was large enough, once a transverse crack initiated in the thick "mud" layer, it was prone to continuous propagation due to the lower fracture toughness of "mud" layer. Thus, an excessive thickness of the "mud" layer would promote the continuous growth of transverse cracks in the model, making the "brick-mud" layered structure TC coating prone to interlayer spalling failure. This was consistent with the experimental results in Figure 9b.

Based on the above analysis, under the combined effect of stress field and the interface height parameters, the trend of transverse cracking during thermal cycling in the "brick-mud" layered structure TC coating increased with the increase in the "mud" layer thickness.

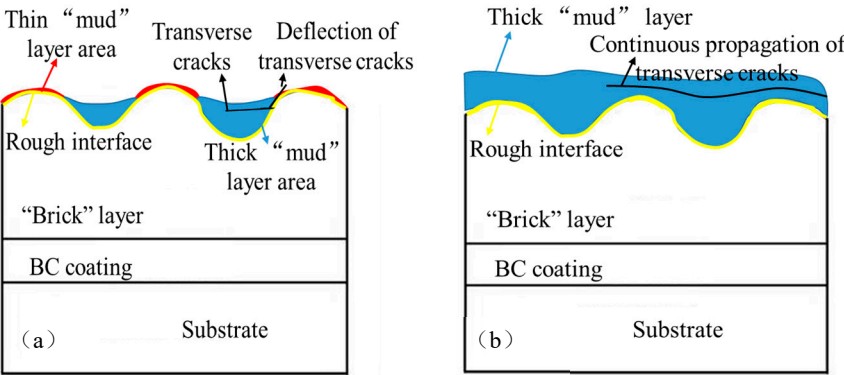

**Figure 23.** Propagation behaviors of transverse cracks in "mud" layers: (**a**) thin "mud" layer; and (**b**) thick "mud" layer.

## 5. Conclusions

(1) As the thickness of the "mud" layer increased, the bonding strength of the "brick-mud" structure TC coatings gradually decreased, and the fracture positions of the tensile specimens were gradually transferred from the TC/BC interface to the "brick" layer/"mud" layer interface.

(2) The thermal cycling lives of the T1 and T2 coatings were improved by 90.0% and 135.7%, respectively, compared with the T0 coating, and their failure modes were overall spalling of the TC coating, while that of the T3 coating decreased by 81.4% with a failure mode of interlayer delamination.

(3) The stress fields of the M2 "mud" layers with different thicknesses were subjected to a comprehensive effect by thermal mismatch stress and pores in the "brick" layer. The thin "mud" layer sustained the largest $\sigma_{22\ max}$, $\sigma_{12\ max}$, and $\sigma_{11\ max}$.

(4) The deflection degree of longitudinal cracks rapidly increased with the increase in the "mud" layer thickness, leading to a significant improvement in its thermal cycling energy dissipation ability, while the propagation length of transverse cracks in the "mud" layer also increased as it increased, and the length of the transverse crack increased sharply when the thickness of "mud" layer increased to 20 μm.

(5) The stress field and the interface height parameters were the main factors affecting the initiation and propagation behavior of thermal cycling cracks in the "brick-mud"-structure TC coating, which, in turn, affected the thermal cycling performance. The TC coating containing thin "mud" layers experienced significant longitudinal cracking and minimal transverse cracking, resulting in an optimal thermal cycling performance.

**Author Contributions:** Methodology, J.W.; Software, L.M.; Validation, Z.W. (Zhibing Wu); Investigation, X.M.; Data curation, Y.D.; Writing—original draft, T.C.; Writing—review & editing, Z.W. (Zhiping Wang) and S.D. All authors have read and agreed to the published version of the manuscript.

**Funding:** This work was supported by Natural Science Research Project of Tianjin Education Commission (Grant No. 2020KJ030) and National Natural Science Foundation of China (Grant No.12202229).

**Institutional Review Board Statement:** Not applicable.

**Informed Consent Statement:** Not applicable.

**Data Availability Statement:** Data are contained within the article.

**Conflicts of Interest:** Author Xiang Ma was employed by the company Aircraft Maintenance and Engineering Corporation Co., Ltd. The remaining authors declare that the research was conducted in the absence of any commercial or financial relationships that could be construed as a potential conflict of interest.

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
