# Peer review of "Experiment and Numerical Simulation on Thermal Cycling Performance of YSZ-Based Sealing Coatings with “Brick-Mud” Layered Structure"

_coatings, doi:10.3390/coatings14030351_

Round 1

Reviewer 1 Report

Comments and Suggestions for Authors

In the paper "Experiment and Numerical Simulation on Thermal Cycling Performance of YSZ-Based Sealing Coatings with "Brick-Mud" Layered Structure" approaches to improve the properties of protective sealing coatings based on YSZ-polymer due to its layer modification are considered. Strength and thermal cycling characteristics are considered and theoretical reconstruction of layer properties is carried out. In general, the work approaches the topic of the journal and can be considered for publication in the journal Coatings after revision of the text according to the comments below.

1. It is necessary to expand the experimental part about the composition of the coating and methods of its preparation. The presence of a reference to previous work is uninformative. It is necessary at least to specify the composition of components, preparation methods and sintering conditions ( Section 2.1.1).

2. It is necessary to describe in more detail the procedure for making layers of "mud" of different thicknesses (Section 2.1.2). As it stands, it is not clear what the substrate is made of and what the choice of distance, etc. is related to.

3. How is the separation of the different layers from the cross-sectional images performed? In the SEM images, the layers are homogeneous in microstructure. Is their chemical analysis different? Can the EDS method fix the difference along the cross-slip of the coating?

4. The title of tables ("Tab") should be given in full "Table"

Author Response

Thank you very much for your helpful suggestions. Please see the attachment.

Reviewer 2 Report

Comments and Suggestions for Authors

A pdf file is attached. 

Comments on the Quality of English Language

Text require polishing

Author Response

(The authors gave the same response as above.)

Reviewer 3 Report

Comments and Suggestions for Authors

Manuscript ID: coatings-2904327

Title: Experiment and numerical simulation on thermal cycling performance of YSZ-based sealing coatings with "brick-mud" layered structure

The manuscript by Cheng Taotao et al. deals with a challenging topic such as fabrication of ceramic sealing coatings for shell bionic structures and the analysis of their thermal performances by numerical simulation.

The manuscript is well written and can be a good contribution to Coatings – MDPI journal.

Nevertheless, the manuscript is lacking in the Introduction section, and it needs to be integrated and improved.

For this reason, the following question should be addressed.

-        The introduction is poor and needs to be integrated with the most recent studies regarding the use of ceramic materials and resins to obtain advanced coatings used in various fields of application (i. e. aerospace, naval, dentistry etc.)

Below are some related articles:

1.     https://doi.org/10.1016/j.jdent.2010.10.015

2.     https://doi.org/10.1016/j.matdes.2014.11.057

3.     https://doi.org/10.1002/app.47481

4.     https://doi.org/10.1021/acsomega.3c01415

5.     https://doi.org/10.1016/j.wear.2007.01.005

In light of the above considerations, MINOR revisions are suggested to integrate the manuscript with further useful details and increase the added value of the contribution.

Comments on the Quality of English Language

Minor editing of English language required

Author Response

(The authors gave the same response as above.)

Reviewer 4 Report

Comments and Suggestions for Authors

Dear Authors,

Thank you very much for your extensive research and detailed analyses.

However, some areas should be improved:

-        Description of the samples is expressed by the maximum surface height (Sz), maximum peak  height (Sp), maximum valley depth (Sv) and arithmetic mean height (Sa). I miss the information about the waviness of the samples or even the filter of form.

-        I completely miss the Aboot-Firestone curve for the surface evaluations. The high viscosity spray gun can generate certain difference in the layers.

-        I miss statistics in your data evaluation. E.g. – you mention average values (with precision in 0.00x micrometers), but no detailed info is about the data, their distributions, standard deviations, or dispersion,

-        Can you clarify the numbers of thermal cycles (Table 6), please? Line 391: When the number of thermal cycles was 60, multiple longitudinal microcracks and smaller transverse cracks appeared in the T2 coating. Was it suddenly or gradually, please?

-        Can you clarify the better understanding of spalling and Peeling from a physical point of view?

-        Figure 3 should be improved

-        Figure 22 – How the “rough” surface is defined, please?

Misprint: Table1  Please, unite: "tempera ture"  to "temperature"

Author Response

(The authors gave the same response as above.)

Round 2

Reviewer 2 Report

Comments and Suggestions for Authors

A revised manuscript can be accepted for publication

Comments on the Quality of English Language

Minor English editing is required